# Artificial Intelligence in Surgical Training for Kidney Cancer: A Systematic Review of the Literature

**DOI:** 10.3390/diagnostics13193070

**Published:** 2023-09-27

**Authors:** Natali Rodriguez Peñaranda, Ahmed Eissa, Stefania Ferretti, Giampaolo Bianchi, Stefano Di Bari, Rui Farinha, Pietro Piazza, Enrico Checcucci, Inés Rivero Belenchón, Alessandro Veccia, Juan Gomez Rivas, Mark Taratkin, Karl-Friedrich Kowalewski, Severin Rodler, Pieter De Backer, Giovanni Enrico Cacciamani, Ruben De Groote, Anthony G. Gallagher, Alexandre Mottrie, Salvatore Micali, Stefano Puliatti

**Affiliations:** 1Department of Urology, Azienda Ospedaliero-Universitaria di Modena, Via Pietro Giardini, 1355, 41126 Baggiovara, Italy; natalirodriguez647@gmail.com (N.R.P.); ahmed.essa@med.tanta.edu.eg (A.E.); ferretti.stefania@aou.mo.it (S.F.); giampaolo.bianchi@unimore.it (G.B.); di.bari.stefano95@gmail.com (S.D.B.); salvatore.micali@unimore.it (S.M.); 2Department of Urology, Faculty of Medicine, Tanta University, Tanta 31527, Egypt; 3Orsi Academy, 9090 Melle, Belgium; ruifarinhaurologia@gmail.com (R.F.); pieter.de.backer@orsi.be (P.D.B.); degroote.ruben@gmail.com (R.D.G.); ag.gallagher@ogcmetrics.com (A.G.G.); a.mottrie@gmail.com (A.M.); 4Urology Department, Lusíadas Hospital, 1500-458 Lisbon, Portugal; 5Division of Urology, IRCCS Azienda Ospedaliero-Universitaria di Bologna, 40138 Bologna, Italy; pietropiazza1209@gmail.com; 6Department of Surgery, FPO-IRCCS Candiolo Cancer Institute, 10060 Turin, Italy; checcu.e@hotmail.it; 7Urology and Nephrology Department, Virgen del Rocío University Hospital, 41013 Seville, Spain; ines.rivero.belenchon@gmail.com; 8Department of Urology, University of Verona, Azienda Ospedaliera Universitaria Integrata, 37126 Verona, Italy; a.veccia88@gmail.com; 9Department of Urology, Hospital Clinico San Carlos, 28040 Madrid, Spain; juangomezr@gmail.com; 10Institute for Urology and Reproductive Health, Sechenov University, 119435 Moscow, Russia; marktaratkin@gmail.com; 11Department of Urology and Urosurgery, University Medical Center Mannheim, Medical Faculty Mannheim, Heidelberg University, 68167 Mannheim, Germany; karl.kowalewski@googlemail.com; 12Department of Urology, University Hospital LMU Munich, 80336 Munich, Germany; severin.rodler@med.uni-muenchen.de; 13Department of Human Structure and Repair, Faculty of Medicine and Health Sciences, Ghent University, 9000 Ghent, Belgium; 14USC Institute of Urology, Catherine and Joseph Aresty Department of Urology, Keck School of Medicine, University of Southern California, Los Angeles, CA 90089, USA; giovanni.cacciamani@gmail.com; 15AI Center at USC Urology, USC Institute of Urology, University of Southern California, Los Angeles, CA 90089, USA; 16Faculty of Life and Health Sciences, Ulster University, Derry BT48 7JL, UK

**Keywords:** RAPN, partial nephrectomy, radical nephrectomy, kidney cancer, renal cancer, annotation, deep learning, computer vision, artificial neural network, artificial intelligence, training, augmented reality, simulation

## Abstract

The prevalence of renal cell carcinoma (RCC) is increasing due to advanced imaging techniques. Surgical resection is the standard treatment, involving complex radical and partial nephrectomy procedures that demand extensive training and planning. Furthermore, artificial intelligence (AI) can potentially aid the training process in the field of kidney cancer. This review explores how artificial intelligence (AI) can create a framework for kidney cancer surgery to address training difficulties. Following PRISMA 2020 criteria, an exhaustive search of PubMed and SCOPUS databases was conducted without any filters or restrictions. Inclusion criteria encompassed original English articles focusing on AI’s role in kidney cancer surgical training. On the other hand, all non-original articles and articles published in any language other than English were excluded. Two independent reviewers assessed the articles, with a third party settling any disagreement. Study specifics, AI tools, methodologies, endpoints, and outcomes were extracted by the same authors. The Oxford Center for Evidence-Based Medicine’s evidence levels were employed to assess the studies. Out of 468 identified records, 14 eligible studies were selected. Potential AI applications in kidney cancer surgical training include analyzing surgical workflow, annotating instruments, identifying tissues, and 3D reconstruction. AI is capable of appraising surgical skills, including the identification of procedural steps and instrument tracking. While AI and augmented reality (AR) enhance training, challenges persist in real-time tracking and registration. The utilization of AI-driven 3D reconstruction proves beneficial for intraoperative guidance and preoperative preparation. Artificial intelligence (AI) shows potential for advancing surgical training by providing unbiased evaluations, personalized feedback, and enhanced learning processes. Yet challenges such as consistent metric measurement, ethical concerns, and data privacy must be addressed. The integration of AI into kidney cancer surgical training offers solutions to training difficulties and a boost to surgical education. However, to fully harness its potential, additional studies are imperative.

## 1. Introduction

Renal cell carcinoma (RCC) represents a common malignancy with approximately 431,288 newly diagnosed cases and 179,368 deaths worldwide in 2020 [1]. Furthermore, the advancements in imaging techniques allow for the detection of tumors at earlier stages [2,3]. However, still, a substantial portion (approximately 10% to 17%) of kidney tumors are classified as benign through histopathological evaluation [4]. Moreover, certain populations with co-existing health conditions, such as obesity and the elderly, face increased risks during interventions. Generally, for non-metastatic disease, surgical resection remains the standard of care. According to the current guidelines, surgery can be performed by either an open, laparoscopic, or robotic-assisted approach dependent on local conditions to maximize oncological, functional, and perioperative outcomes [5].

Partial and complex radical nephrectomy are challenging procedures that require thorough training and planning for the key steps such as dissection of the renal hilum, tumor enucleation, and renorraphy. However, due to the triple burden of patient care, research, and teaching, especially for doctors in academic centers, exposure to hands-on experience in the operating room (OR) and surgical training have become less prominent in current curricula [6]. Thus, training outside the OR in various simulation scenarios has become more important. Surgical training outside the OR can be provided, for example, by dry lab laparoscopy training [7,8], virtual reality training [9], or serious gaming [10]. All these modalities have been shown to be beneficial for the trainees.

In addition to spare resources even during training, the recent advances in technology might help to replace trainers with training systems that can assess trainees’ performance based on metric parameters [11]. Furthermore, it has been shown that artificial intelligence (AI) may greatly contribute to the improvement and automation of surgical training [12,13]. 

In 1955, Professor John McCarthy of Stanford University coined the term artificial intelligence for the first time, referring to the capability of building intelligent machines that can efficiently perform complex intellectual human tasks such as learning, thinking, reasoning, and problem-solving [14]. Generally, AI depends on the quantity, quality, and variability of the available data for the training of these models and systems, which is considered one of the major challenges to the development and robustness of different AI applications [15,16]. The potential advantages of this technology fascinated the health care industry, allowing its permeation in nearly all fields of medicine. This increasing interest in AI applications in the medical field was further aided by the advancements in medical technology, such as the shift to electronic medical records, digital radiology, digital pathology, and minimally invasive surgeries such as robotics, laparoscopic surgery, and endoscopy, which allowed the generation of large amounts of data [15]. 

AI in the healthcare industry consists of four subfields. Firstly, machine learning (ML) defines the use of dynamic algorithms to identify and learn from complex patterns in a data set, thus allowing the machine to make accurate predictions. Secondly, Natural Language Processing (NLP) is another subfield of AI that encompasses the ability of the computer to understand and process written and spoken languages [17]. Thirdly, deep learning (DL) includes the use of massive datasets to train individualized functioning units, which are arranged in multiple connected layers resembling artificial neurons. These functioning units are known as artificial neural networks (ANN) [18]. Finally, computer vision (CV) is the ability of the computer to identify and analyze different objects in an image or a video [17].

In these settings, the current systematic review aims to assess how AI might help to overcome the current limitations of surgical education and to establish a dedicated framework for kidney cancer surgery, which proves particularly intricate, especially in the tumor enucleation and renorraphy steps (two surgical aspects where intelligence can be immensely beneficial).

## 2. Materials and Methods

### 2.1. Search Strategy

A systematic review of the literature was carried out in accordance with the Preferred Reporting Items for Systematic Reviews and Meta-Analysis (PRISMA) statement [19]. Furthermore, the PRISMA 2020 checklist (Appendix A), PRISMA Abstract checklist (Appendix A) [19], and PRISMA-S checklist (Appendix A) [20] are added as Appendix A. A systematic search of the PubMed and SCOPUS databases was conducted on the 7 August 2023, to identify all articles concerned with the use of AI in kidney cancer surgical training. A combination of the following terms was used for the search: “robotic assisted partial nephrectomy”, “RAPN”, “partial nephrectomy”, “radical nephrectomy”, “nephroureterectomy”, “kidney cancer”, “Renal cancer”, “Annotation”, “machine learning”, “Deep learning”, “natural language processing”, “computer vision”, “artificial neural network”, “artificial intelligence”, “CV”, “NLP”, “DL”, “ANN”, “ML”, “AI”, “training”, “performance assessment”, “performance evaluation”, “virtual reality”, “VR”, “augmented reality”, “AR”, “simulation”, and “workflow”. Appendix A shows the combination of the keywords used for each database searched. 

### 2.2. Search Criteria

The inclusion criteria consisted of all original articles focusing on the utility of AI for surgical training of kidney cancer without any restrictions on the type of study (retrospective or prospective case series, clinical trials, cohort studies, or randomized controlled trials) or the date of publication. The articles were excluded if they were not published in the English language, had no original data (reviews, letters to the editor, commentaries, and editorials), or the full text was not available.

### 2.3. Screening and Article Selection

Two independent authors (AE and NR) screened all the search results by title and abstract to identify all the articles with clinical relevance to the topic of the current review according to the predefined inclusion and exclusion criteria. Duplicates were examined using Mendeley reference manager (Elsevier Ltd., Amsterdam, The Netherlands) and were revised manually for exclusion. Subsequently, a full-text review was performed for all the remaining manuscripts after the initial screening to finally determine the articles that will be discussed in the current review. A manual review of the references in the included studies was performed to identify any relevant articles. Finally, a third author (SP) reviewed the search process and helped in resolving any discrepancies among the two reviewers.

### 2.4. Data Extraction

Data from the determined articles was collected independently by the same two authors in a standard Excel sheet. The following key aspects of the included studies were extracted: (1) first author and year of publication; (2) type of the study; (3) number of patients or cases included; (4) AI tools used for building the algorithms of the study; (5) a brief description of the methods used; (6) the main endpoints; and (7) the main findings of the study. 

### 2.5. Level of Evidence

Finally, the included studies were evaluated according to the 2011 Oxford Center for Evidence-Based Medicine (OCEBM) level of evidence [21]. OCEBM levels function as a hierarchical guide to identify the most reliable evidence. They are designed to offer a quick reference for busy clinicians, researchers, or patients seeking the strongest available evidence; the OCEBM Levels aid clinicians in swiftly appraising evidence on their own. While pre-appraised sources such as Clinical Evidence, NHS Clinical Knowledge Summaries, and UpToDate may offer more extensive information, they carry the potential for over-reliance on expert opinion. Additionally, it’s important to note that the OCEBM levels do not provide a definitive assessment of evidence quality. In certain instances, lower-level evidence, such as a compelling observational study, can yield stronger evidence compared to a higher-level study, such as a systematic review with inconclusive findings. Moreover, the levels do not offer specific recommendations; they act as a framework for evaluating evidence, and ultimate decisions should be guided by clinical judgment and the unique circumstances of each patient. In short, the levels serve as efficient tools for swift clinical decision-making, eliminating the reliance on pre-appraised sources. They provide practical rules of thumb comparable in effectiveness to more intricate approaches. Importantly, they encompass a broad spectrum of clinical questions, enabling the assessment of evidence regarding prevalence, diagnostic accuracy, prognosis, treatment effects, risks, and screening effectiveness [22].

## 3. Results

### 3.1. Search Results

Overall, the search identified 468 records, of which 53 articles were excluded as they were duplicates. The initial screening of the remaining 415 articles by title and abstract resulted in the exclusion of 385 articles that did not meet the inclusion criteria of the current review. The remaining 30 articles were eligible for full-text review, after which another 16 were excluded for different reasons. Only one article caused disagreement among the reviewers, where the authors assessed the validity of an AI total that may guide the decision to either perform partial or radical nephrectomy (after discussion among the reviewers, it was excluded as this tool has no effect on the surgical skills of the novice) [23]. Finally, 14 articles met our inclusion criteria and were included in the current systematic review [24,25,26,27,28,29,30,31,32,33,34,35,36,37]. Figure 1 illustrates the PRISMA flow diagram for the search process.

### 3.2. AI and Surgical Training for Kidney Cancer

There is a scarcity of evidence in the literature regarding the application of AI to enhance surgical skills in the urological discipline. This also applies to the field of surgical training for kidney cancer, where AI has predominantly been employed in the realm of surgical simulations and robot-assisted surgery [24,25,26,27,28,29,30,31,32,33,34,35,36,37]. Table 1 shows a summary of the included studies concerning the use of AI in the field of renal cancer training.

#### 3.2.1. AI and Performance Assessment

The application of AI techniques in the automated analysis of surgical video is crucial and holds central importance in this context. Subfields, such as CV and ML, are increasingly being applied to surgical videos, enabling surgical workflow analysis, intraoperative guidance, and objective assessment of actions, errors, and risks. Detecting and estimating the pose and movement of surgical instruments plays a key role in surgical video and image analysis for performance assessment [32]. It can also provide insights into the surgeon’s intent and facilitate a better understanding of the surgical workflow [26]. In this setting, Nakawala et al., used DL algorithms to obtain a detailed workflow of RAPN by implementing a “Deep-Onto” network on surgical videos, which accurately (74.3%) identified not only the steps of RAPN but also anatomy and instruments [24]. This “Deep-Onto” network consisted mainly of two models, the first of which is made of two components: a convolutional recurrent neural network (CRNN) model that is responsible for the recognition of the current surgical step and a “Sequence” model that uses the outcomes provided by the CRNN model to anticipate the upcoming surgical step. Secondly, a “Knowledge” model is applied to provide further information about the ongoing step including the instruments used, actions, and phase [24]. The same authors combined the predictions of the CRNN and “Subsequent” models as a predicate defining the ongoing surgical step, and the corresponding consecutive step, where correctly predicted step sequences were identified as positive examples in the Inductive Logic Programming (ILP) system. Subsequently, ILP was applied to identify relational information between different surgical entities [26]. 

However, instrument annotation represents the cornerstone for robotic surgical AI projects focused on instrument detection and surgical workflow analysis; no clear guidelines for instrument annotation are currently available. In this setting, Pieter De Backer et al. [35] have developed an efficient bottom-up approach for team annotation of robotic instruments in robot-assisted partial nephrectomy (RAPN). Interestingly, a recent study used 872 labeled images from 150 segmented RAPN videos to train a multi-task convolutional neural network model to predict the surgical proficiency scores as Objective Structured Assessment of Technical Skills (OSATS) and Global Evaluative Assessment of Robotic Skills (GEARS), demonstrating that the model’s performance was comparable to human ratters in some subcategories such as identification of instruments and force sensitivity; however, it may be generally associated with less reliable results compared to human ratters. The authors explained these findings by citing the lack of data that was used for model training [32]. Despite the limitations reported in this study, it represents a progression for automated performance assessment in the field of renal cancer. 

#### 3.2.2. AI and AR

On the other hand, the integration of virtual reality and augmented reality with AI is capable of improving minimally invasive renal cancer surgery training, planning, and outcomes. In this setting, the ORSI Academy workgroup showcased the potential for enhancing augmented reality in robotic renal surgery through DL techniques [33]. They developed an algorithm based on deep learning networks to identify nonorganic objects, specifically robotic instruments, during robot-assisted partial nephrectomy and renal transplantation. This enabled the overlay of virtual 3-D images onto the surgical stream without obstructing the view of the robotic tools. As a result, surgeons could employ the tools while simultaneously benefiting from this augmented virtual reality approach. Although the study encountered some challenges, it is undeniably an encouraging step forward in the field [33]. Similarly, Padovan E, et al., proposed a framework consisting of a segmentation CNN that differentiates between structures in the endoscopic view and a rotation CNN that calculates the rotation values (X, Y, and Z axis) [25]. The performance and function of the rotation CNN differ according to the anchoring structure; in the case of an anchoring rigid instrument such as urethral catheter in patients undergoing robotic-assisted radical prostatectomy, it can be used for both registration and tracking tasks. On the other hand, a soft tissue anchoring structure can be more complex as a result of its deformability; thus, rotation CNN can be used only for registration (as in the prostate), and it can even require manual adjustments in the case of more complex organs (such as the kidneys) [25]. Considering laparoscopic renal surgery, an automatic deformable marker-less registration framework of a video see-through AR system demonstrated an average registration error of 1.28 mm when tested on renal phantom models [29]. This system is based on a disparity map of the laparoscopic image that is generated using a semi-global block matching method. The disparity image is used for accurate pixel-to-pixel matching through path-wise optimization of the global cost function. Subsequently, a manually trained Mask R-CNN is used for segmentation of the renal surface regions. Finally, 3D reconstruction within the identified region is performed to attain the reconstruction point cloud of the renal surface [29].

In line with the previous studies, CV in the form of a modified Center Surrounded Extremas for Real-time Feature Detection (Cen-SuRE) known as STAR in combination with Binary Robust Independent Elementary Feature (BRIEF) was successfully used to detect salient features in an endoscopic frame that can be employed for real-time tissue tracking during RAPN [36]. Moreover, Amparore et al., used CV technology to aid the automatic anchoring of the 3D model over the real anatomy during RAPN. The authors differentiated the kidney from surrounding structures using super-enhancement of the kidney by employing indocyanine green (ICG). Despite the promising results reported by the authors, there were some limitations, including the need for human interference during the initial phases, the failure of anchoring in cases of posterior lesions where the kidney is largely rotated, and the burdensome tracking when there was not homogenous ICG perfusion [34].

Hilar dissection and clamping of renal arteries are among the most demanding steps of partial nephrectomy, particularly for novice surgeons, as any missed vessel or improper clamping may result in significant bleeding and compromise the outcomes of surgery. In these settings, different AI algorithms, such as the 3D Fully Convolutional Neural Network, a modified Eulerian motion processing technique, have been applied for the segmentation of renal vessels and the identification of occluded vessels that might not be visible to the naked eye during minimally invasive partial nephrectomy [27,30,37]. Similarly, Nosarti MS et al. [31] utilized color and texture visual cues through the use of random decision forest algorithms together with the preoperative data obtained from imaging studies for segmentation of the endoscopic view that may help in the identification of occluded tissues such as vessels occluded by fat and endophytic tumors. Furthermore, it can be used for augmentation of endoscopic view through tissue tracking [31].

#### 3.2.3. AI and 3D-Printing

On the other hand, DL algorithms were proposed for the 3D reconstruction of renal models in patients with giant angiomyolipomas, which in turn can be printed to help in preoperative planning, patient-specific rehearsal, and intraoperative guidance during even open surgery. This combination of 3D printing and AI has the potential to enhance training and improve surgical outcomes [28]. In comparison with standard techniques, this combination resulted in significantly higher success rates for partial nephrectomy (30% versus 72%) [28].

## 4. Discussion

AI will potentially change the landscape of medicine and reshape the healthcare industry over the coming years. Generally, the applications of AI in healthcare include, but are not limited to, drug development, health monitoring, medical data management, disease diagnostics and decision aids, digital consultation, personalized disease treatment, analysis of health plans, and surgical education [38,39,40]. Expectations for AI applications in medicine are high, and some workers in the healthcare industry believe that if AI systems are currently capable of efficiently driving cars, they might be able to autonomously control surgical robots one day. However, it should be noted that AI is not a replacement for the human factor; it is just a tool to help medical professionals do their job more efficiently and safely [41]. The urological field is not an exception, where AI has made significant advancements in enhancing diagnosis, prognosis, outcome prediction, and treatment planning [17,42,43,44,45].

Considering surgical training, numerous studies have emphasized the significance of surgical skills in determining patient outcomes, including mortality, complication rates, operation length, and re-operation and re-admission rates [46,47]. Interestingly, surgical skills may account for up to 25% of the variation in patient outcomes [48]. Therefore, it is crucial to evaluate surgical skills effectively to enhance training, credentialing, and education and ultimately provide the highest quality of care to patients. 

Medical training has traditionally followed the Halstedian model, where trainees observe, perform, and then teach procedures [49]. However, new regulations, increased paperwork, and concerns about inexperienced surgeons operating on patients have highlighted the need for a change in surgical training [49,50]. This is particularly important because studies have shown that complications tend to occur during the early stages of a surgeon’s learning curve [51]. Therefore, surgical training should prioritize structured and validated processes, including proficiency-based progression training and objective assessments. Accordingly, training should involve practice in a controlled setting or through simulations, where trainees could only move on to real-life procedures after reaching an established proficiency benchmark to enhance patient safety [52,53]. Noteworthy, the process of performance evaluation requires manual peer appraisal by trained surgical experts either during surgery or review of surgical videos. This is a time-consuming and unreliable process due to the lack of a standardized definition of success among different surgeons [54]. 

Interestingly, AI can be integrated with conventional proficiency-based training approaches to provide an objective assessment of surgical skills [52], which has paved the way for the development of automated machine-based scoring methods, particularly in the field of robotic surgery [55,56,57,58]. A recent randomized control trial demonstrated that the virtual operative assistant system (an AI-based tutoring system) provided superior performance outcomes and better skill acquisition compared to remote expert tutoring [59].

Considering autonomous assessment of surgical skills during kidney cancer surgeries, we are still taking our first steps, where most of the published studies are concerned with the identification of surgical workflow of RAPN [24,26], instrument annotation [33,35], and different tissue identification (blood vessels, tumors, anatomical spaces) [27,30,31,37], while only one study took a step forward towards actual automatic assessment of surgical skills during RAPN [32].

In this context, it’s evident how kidney imaging plays a pivotal role in the field of AI applied to renal surgical training. Over recent years, there has been a consistent drive for innovative imaging techniques. These developments aim to empower surgeons to conduct thorough examinations of the kidney, including remarkable three-dimensional (3D) reconstruction technology. The use of 3D models results in patient-specific virtual or physical replicas, leading to reduced operative time, clamping duration, and estimated blood loss [60,61]. Additionally, these techniques play a crucial role in educating patients and their families, providing a deeper understanding of tumor characteristics and the range of available treatment options [62,63]. The application of 3D-volumetry, a technology utilizing CT scans to assess renal volume, proves essential in evaluating split renal function [64]. The introduction of holographic technology represents a ground-breaking approach, providing an immersive and interactive experience based on 3D visualization. This fosters a greater appreciation of patient-specific anatomy [65].

Moreover, intraoperative navigation guided by 3D virtual models leads to lower complication rates and improved outcomes [66]. Intraoperative imaging, which encompasses both morphological and fluorescence techniques, is vital for tumor identification and assessment of ischemia through tools such as laparoscopic US probes [67] and intraoperative fluorescent imaging [68]. Embracing pathological intraoperative imaging through technologies such as fluorescence confocal microscopy [69,70] and optical coherence tomography [71] holds immense promise in guiding the treatment of small renal masses and advancing cancer control.

In their comprehensive examination of innovative imaging technologies for robotic kidney cancer surgery, Puliatti et al., meticulously detail the techniques mentioned above, highlighting the potential application of 3D visualization technologies and augmented reality navigation for guiding operations and providing training in renal cancer surgery [72]. Furthermore, the integration of VR and AR with AI is capable of improving robotic renal cancer surgeries and training [73]. Particularly, AR may have a great potential for reducing surgical complications and improving outcomes after surgery by guiding novice surgeons through the initial learning curve. However, the main limitation of AR-guided surgery is the registration process, where the 3D reconstructed model is superimposed over the corresponding anatomy in the endoscopic view. Furthermore, real-time object or structure tracking is another concern. Particularly, the kidneys are more complex organs (as they are not fixed to anatomical constraints) compared to other ones, such as the prostate, and thus require more sophisticated registration and tracking techniques [25]. In these settings, four studies in the literature focused on the use of AI for the automatic registration of 3D models and for real-time tissue tracking during surgery to overcome the limitations of tissue deformability and mobility [25,29,34,36]. Future work should concentrate on annotating soft tissues to study and quantify tool-tissue interactions. Accurate soft tissue segmentation in conjunction with instrument segmentation is crucial for successful augmented reality applications, ensuring the correct registration of 3D models with the intraoperative view [74].

In these settings, training machines to identify anatomical spaces, instruments, and different stages of various procedures is crucial not just for offering real-time assistance and feedback to surgeons during operations but also for providing young surgeons in a training environment with an understanding of their proficiency level in a specific surgical step. When it comes to kidney cancer, we still have a considerable distance to go before achieving this type of application.

### 4.1. AI Limitations

One of the major limitations restricting AI-based publications in the field of kidney cancer surgical training is the lack of standardized metrics for the objective evaluation of trainees’ ability to perform RAPN. Thus, Rui Farinha et al. [75] presented an international expert consensus on the metric-based characterization of left-sided RAPN cases, but these metrics are not applicable to other scenarios. They also established evidence supporting the validity of these metrics, showing reliable scoring and discrimination between experienced and novice RAPN surgeons [76]. However, the progression in the field of RAPN metrics development in the near future will allow the integration of AI to enable real-time error recognition during surgery and provide feedback during training, which holds promise for offering valuable insights and assistance in enhancing surgical performance.

Additionally, robotic surgery serves as an ideal testing ground for the advancement of AI-based programs due to its ability to capture detailed records of surgeons’ movements and provide continuous visualization of instruments. According to a Delphi consensus statement in 2022, the integration of AI into robotic surgical training holds significant promise, but it also introduces ethical risks. These risks encompass data privacy, transparency, biases, accountability, and liabilities that require recognition and resolution [77].

Furthermore, despite the extensive published literature on the significant potential of AI, there are no reports on its efficacy in improving patient safety in robot-assisted surgery [14].

Other general limitations to the robustness and acceptance of AI applications in the field of medicine include the heterogeneity of the research methodology of the published studies, the restricted generalizability as most of the developed AI algorithms are trained and validated using similar datasets, which may result in overfitting of the models, and the fact that many of the ML algorithms (particularly ANN) are very complex and difficult to understand, resembling a black box, which represents an obstacle towards rigorous testing of these algorithms [78]. Finally, the availability of large labeled datasets in the medical field is scarce in comparison to other fields [79]. In this setting, verified benchmark datasets are essential; however, producing a high-quality benchmark dataset is a complex and time-consuming process [80]. Most of the available benchmark datasets in the field of kidney cancer are mainly related to the histopathological or radiological evaluation of kidneys (i.e., the 2019 kidney and kidney tumor segmentation challenge [KiTS19], KiTS21, and KMC datasets) [81,82,83]. According to our knowledge, there are no benchmark datasets related to the segmentation of instruments and tissues during minimally invasive partial nephrectomy. Noteworthy, it is not always about the size and quality of the dataset. Gaël Varoquaux and colleagues effectively shed light on methodological errors in various aspects of clinical imaging within their review [79]. Despite extensive research in this field, the clinical impact remains constrained. The review pinpoints challenges in dataset selection, evaluation methods, and publication incentives, proposing strategies for improvement. It underscores the necessity for procedural and normative changes to unlock the full potential of machine learning in healthcare. The authors stress the tendency for research to be influenced by academic incentives rather than meeting the needs of clinicians and patients. Dataset bias is identified as a significant concern, emphasizing the importance of accurately representative datasets. Robust evaluation methods beyond benchmark performance are called for, along with the adoption of sound statistical practices. The publication process may fall short of promoting clarity, potentially hindering reproducibility and transparency. Researchers are urged to prioritize scientific problem-solving over publication optimization, considering broader impacts beyond benchmarks [79].

### 4.2. Future Perspectives

AI-powered technologies will impact all areas of surgical education. Starting with the communication between trainers and trainees, AI-powered language translators will allow an English trainer/trainee and a Chinese-speaking trainee/trainer, to interact in their native languages, improving communication and breaking language/cultural barriers. Moving to e-learning platforms, where AI can be used to fully guarantee trainees’ identification through biometric facial recognition technologies, thus allowing personalized e-learning courses that are customized to the previous knowledge and skills of each trainee. Furthermore, AI will facilitate the classification, editing, and tagging of videos to create personalized e-learning platform content. 

Soon, AI-tutoring systems will be able to objectively assess trainees’ surgical skills in the laboratory and clinical practice, providing personalized quantitative feedback on their technical proficiency. This feedback will help identify areas for skill improvement and track their progress over time. Defining expert proficiency benchmarks will become easier, and comparing them with the trainee’s performance will facilitate proficiency-based training. This monitoring will allow an active and constant adaptation of the training program based on individual needs. These systems will use CV and ML techniques to monitor trainees’ actions, identify errors and critical errors, and offer corrective advice. They will serve as virtual trainers, enhancing the learning experience and ensuring best practices are followed. A bilateral conversation, activated through voice recognition software, will allow personalized virtual trainers to answer questions and keep trainees on track of their progress.

Furthermore, combining AI with VR and AR technologies will increase the potential to create impactful and immersive education and training experiences. This combination will be capable of guiding novice surgeons step-by-step during their initial learning curve, thus improving surgical outcomes.

Finally, AI will aid in continuous learning and knowledge integration since its algorithms will process vast amounts of medical literature, surgical videos, and patient data, extracting relevant insights, identifying trends, and providing trainees with up-to-date information.

## 5. Conclusions

The applications of AI in the field of surgical training for kidney cancer are still in the initial phase of discovery, with multiple limitations and restrictions. However, AI-based surgical training holds the promise of improving the quality of surgical training without compromising patients’ safety. Further studies are required to explore the potential of this technology in the surgical education of renal cancer. 

## Figures and Tables

**Figure 1 diagnostics-13-03070-f001:**
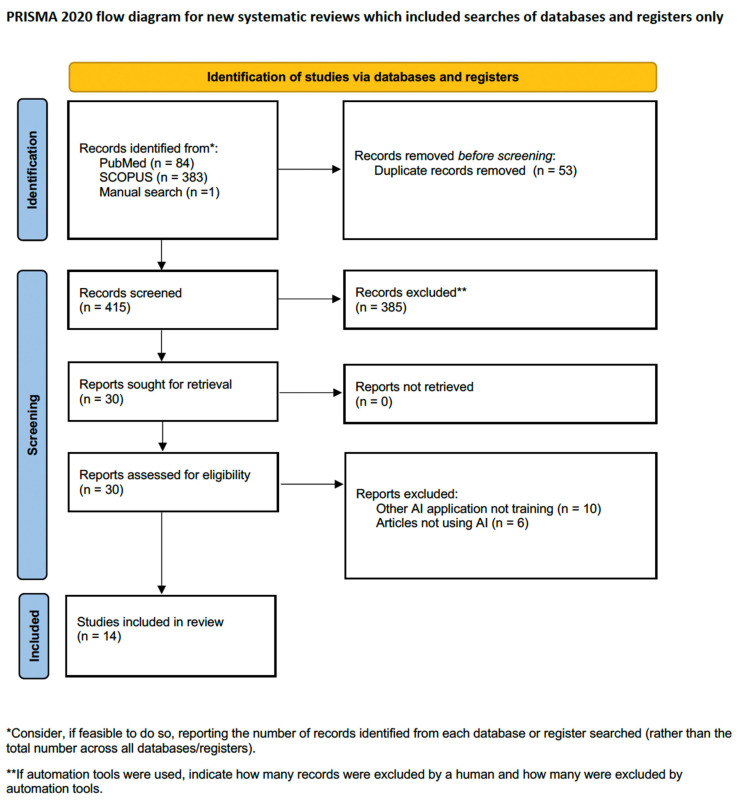
PRISMA flow diagram for the search process.

**Table 1 diagnostics-13-03070-t001:** Summary of the included studies concerning the use of AI in the field of renal cancer training.

Reference	N. of Cases	AI Tool	Study Summary	LE
Instruments and objects annotation/segmentation				
Amir-Khalili A, et al., 2014 [30]	Dataset obtained from eight RAPN videos	Computer vision	Using phase-based video magnification for automated identification of faint motion invisible to human eyes (resulting from small blood vessels hidden within the fat around the renal hilum), thus facilitating the identification of renal vessels during surgery.	5
Amir-Khalili A, et al., 2015 [37]	Validation in 15 RAPN videos	Computer vision	The authors used color and texture visual cues in combination with pulsatile motion to automatically identify renal vessels (especially those concealed by fat) during minimally invasive partial nephrectomy (an extension of their previous study [27]). The area under the ROC curve for this technique was 0.72.	5
Nosrati M. S, et al., 2016 [31]	Dataset obtained from 15 RAPN videos	Machine learning (random decision forest)	Using data from the preoperative imaging studies (to estimate the 3D pose and deformities of anatomical structures) together with color and texture visual cues (obtained from the endoscopic field) for automatic real-time tissue tracking and identification of occluded structures. This technique improved structure identification by 45%.	5
Nakawala H, et al., 2019 [24]	9 RAPN	Deep learning (CRNN and CNN-HMM)	The authors used a dataset obtained from splitting nine RAPN videos into small videos of 30 s and annotated them to train the Deep-Onto model to identify the surgical workflow of RAPN. The model resulted in the identification of 10 RAPN steps with an accuracy of 74.29%.	5
Nakawala H, et al., 2020 [26]	9 RAPN	Deep learning (CNN and LTSM and ILP)	The authors extended their previous work [21] on automatic analysis of the surgical workflow of RAPN. The authors successfully introduced new AI algorithms to use relational information between surgical entities to predict the current surgical step and the corresponding surgical step.	5
Casella A, et al., 2020 [27]	8 RAPN videos	Deep learning (Fully CNNs)	The authors trained a 3D Fully convolutional neural network using 741,573 frames extracted from eight RAPN for segmentation of renal vessels. Subsequently, 240 frames were used for validation, and the last 240 frames were used for testing the algorithm.	5
De Backer P, et al., 2022 [35]	82 videos of RAPN	Deep learning	The authors presented a “bottom-up” framework for the annotation of surgical instruments in RAPN. Subsequently, the images annotated using this framework were validated for use by a deep learning algorithm.	5
De Backer P, et al., 2023 [33]	10 RAPN patients	Deep learning (ANN)	65,927 labeled instruments in 15,100 video frames obtained from 57 RAPN videos were used to train an ANN model to correctly annotate surgical instruments. This model aims for real-time identification of robotic instruments during augmented reality-guided RAPN.	5
**Tissue tracking and 3D registration/augmented reality**				
Yip M.C., et al., 2012 [36]	Laparoscopic partial nephrectomy videos	Computer vision	Using different methods of computer vision for real-time tissue tracking during minimally invasive surgeries, both in vitro (porcine models) and in vivo (laparoscopic partial nephrectomy videos), in order to provide more accurate registration of 3D models.	5
Zhang X, et al., 2019 [29]	1062 images from 9 different laparoscopic partial nephrectomy	Computer vision and Deep learning (CNN)	Computer vision and machine learning were used to develop an automatic markerless, deformable registration framework for laparoscopic partial nephrectomy. The proposed technique was able to provide automatic segmentation with an accuracy of 94.4%.	5
Gao Y, et al., 2021 [28]	31 patients with Giant RAML	Deep learning	The authors used deep learning segmentation algorithms to aid in the creation of 3D-printed models of patients with giant renal angiomyolipoma. Subsequently, they compared surgeries performed with the aid of deep learning—3D models—versus standard surgeries (without the aid of models). Deep learning—3D-printed models resulted in higher rates of partial nephrectomy compared to routine surgeries.	3b
Padovan E, et al., 2022 [25]	9 RAPN videos	Deep Learning (CNN)	The segmentation CNN model uses the RGB images from the endoscopic view to differentiate between different structures in the field. Subsequently, a rotation CNN model is used to calculate the rotation values based on a rigid instrument or an organ. Finally, the information from both models is used for automatic registration and orientation of the 3D model over the endoscopic field during RAPN. This model showed good performance.	5
Amparore D, et al., 2022 [34]	10 RAPN patients	Computer vision	During surgery, super-enhancement of the kidney using ICG was performed to differentiate it from surrounding structures in the firefly mode of the DaVinci robot. Registration of the AR model was performed using the kidney as an anchoring site (fine tuning can be performed by a professional operator), followed by automatic anchoring during the surgery. This technique was successfully applied for superimposing the 3D model over the kidney during seven RAPN cases with completely endophytic tumors.	5
**Skills assessment**				
Wang Y, et al., 2023 [32]	872 images from 150 segmented RAPN videos	Deep learning (multi-task CNN)	Segmented RAPN videos were initially evaluated by human reviewers using GEARS and OSATS scores. Subsequently, a human reviewer labeled each portion of the robotic instruments in a subset of videos. These labels were used to train the semantic segmentation network. A multi-task CNN was used to predict the GEARS and OSATS scores. The model performed well in terms of prediction of force sensitivity and knowledge of instruments, but further training of the model is required to improve its overall evaluation of surgical skills.	5

N. = number; AI = artificial intelligence; CRNN = convolutional recurrent neural network; CNN-HMM = convolutional neural network with hidden Markov Models; ICG = indocyanine green; CNN = convolutional neural network; GEARS = Global Evaluative Assessment of Robotic Skills; OSATS = Objective Assessment of Technical Skills; ANN = artificial neural network; RAML = Renal Angiomyolipoma; ROC = Receiver Operating Characteristic Curve; RAPN = robotic-assisted partial nephrectomy.

## Data Availability

Not applicable.

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
