# Peer review of "Artificial Intelligence in Surgical Training for Kidney Cancer: A Systematic Review of the Literature"

_diagnostics, 2023, doi:10.3390/diagnostics13193070_

Round 1
Reviewer 1 Report
I would like to congratulate the authors for their work. AI in surgery is a very "hot" topic, especially when robotic surgery is considered.
I only have one comment and one observation to address to the authors.
1. You mentioned in the introduction that radical and partial nephrectomy are 2 complex surgeries that require extensive preparation. In my opinion, partial nephrectomy is a complex surgery due to the excision and rrenorraphy part, whereas radical surgery for the kidney is not as complicated. I would highlight that the tumor excision and rrenoraphy part are the most complex parts of kidney cancer surgery and can benefit the most from the aid of AI.
2. Table 1 is very difficult to read; too long. Please revise.
Thank you and good luck in publishing this work!
-
Author Response
Thank you very much for taking the time to review this manuscript. Please find the detailed responses below and the corresponding revision changes in the re-submitted files.
Reviewer 1:
I would like to congratulate the authors for their work. AI in surgery is a very "hot" topic, especially when robotic surgery is considered.
I only have one comment and one observation to address to the authors.
Thanks for your nice comment we really appreciate it.
- You mentioned in the introduction that radical and partial nephrectomy are 2 complex surgeries that require extensive preparation. In my opinion, partial nephrectomy is a complex surgery due to the excision and rrenorraphy part, whereas radical surgery for the kidney is not as complicated. I would highlight that the tumor excision and rrenoraphy part are the most complex parts of kidney cancer surgery and can benefit the most from the aid of AI.
We modified this part in the introduction to focus mainly on partial nephrectomy and just complex cases of radical nephrectomy. Furthermore, we modified the aim of the study to reflect your opinion as follows:
“In these settings, the current systematic review aims to assess how AI might help to overcome the current limitations of surgical education and to establish a dedicated framework for kidney cancer surgery, which proves particularly intricate, especially in the tumor enucleation and renorraphy steps (two surgical aspects where intelligence can be immensely beneficial).”
- Table 1 is very difficult to read; too long. Please revise.
Thank you and good luck in publishing this work!
We understand your concern, we simplified the table to make it easier for the reader.
As regards language
The article was revised as regards the language.
Reviewer 2 Report
The authors have conducted systematic literature review under PRISMA guidelines, covering how AI can improve image recognition with emphasis on kidney cancer surgery. Following are few suggestions to improve the manuscript:
1. Brief explanation of evaluation criteria such as 2011 Oxford Center for evidence-based medicine level of evidence would provide a refresher for new readers.
2. Summarizing any benchmark datasets available in the field would be helpful for the readers.
3. The manuscript fails to summarize the challenges such as explainability of models and the state of art for the field of image diagnosis in medical, with the specialization of kidney cancer. Such as following references guide towards use of AI in image diagnosis in medical field.
Varoquaux, G., Cheplygina, V. Machine learning for medical imaging: methodological failures and recommendations for the future. npj Digit. Med. 5, 48 (2022). https://doi.org/10.1038/s41746-022-00592-y
Ziatdinov, M., Ghosh, A., Wong, C.Y.(. et al. AtomAI framework for deep learning analysis of image and spectroscopy data in electron and scanning probe microscopy. Nat Mach Intell 4, 1101–1112 (2022). https://doi.org/10.1038/s42256-022-00555-8
Hou, J., Gao, T. Explainable DCNN based chest X-ray image analysis and classification for COVID-19 pneumonia detection. Sci Rep 11, 16071 (2021). https://doi.org/10.1038/s41598-021-95680-6
Author Response
Reply to reviewers
Thank you very much for taking the time to review this manuscript. Please find the detailed responses below and the corresponding revision changes in the re-submitted files.
Reviewer 2:
The authors have conducted systematic literature review under PRISMA guidelines, covering how AI can improve image recognition with emphasis on kidney cancer surgery. Following are few suggestions to improve the manuscript:
Thanks for your comment.
- Brief explanation of evaluation criteria such as 2011 Oxford Center for evidence-based medicine level of evidence would provide a refresher for new readers.
Thanks for your comment, we understand your concern, we have added explanation of the OCEBM in the subsection of the level of evidence as per your suggestion as follows:
“Finally, the included studies were evaluated according to 2011 Oxford Center for Evidence-Based Medicine (OCEBM) level of evidence [21]. OCEBM levels function as a hierarchical guide to identify the most reliable evidence. It’s designed to offer a quick reference for busy clinicians, researchers, or patients seeking the strongest available evidence; the OCEBM Levels aids clinicians in swiftly appraising evidence on their own. While pre-appraised sources like Clinical Evidence, NHS Clinical Knowledge Summaries, and UpToDate may offer more extensive information, they carry the potential for over-reliance on expert opinion. Additionally, it's important to note that the OCEBM Levels do not provide a definitive assessment of evidence quality. In certain instances, lower-level evidence, such as a compelling observational study, can yield stronger evidence compared to a higher-level study, like a systematic review with inconclusive findings. Moreover, the Levels do not offer specific recommendations; they act as a framework for evaluating evidence, and ultimate decisions should be guided by clinical judgment and the unique circumstances of each patient. In short, the levels serve as efficient tools for swift clinical decision-making, eliminating the reliance on pre-appraised sources. They provide practical rules of thumb, comparable in effectiveness to more intricate approaches. Importantly, they encompass a broad spectrum of clinical questions, enabling assessment of evidence regarding prevalence, diagnostic accuracy, prognosis, treatment effects, risks, and screening effectiveness [22].”
- Summarizing any benchmark datasets available in the field would be helpful for the readers.
Thanks for your comment, we appreciate your modification. We added a short paragraph about benchmark datasets; however, we made it short just because most of the available benchmark datasets are out of our focus.., mainly concentrating on histopathological and radiological evaluation of kidney cancer
“Finally, the availability of large labelled datasets in the medical field is scarce in comparison to other fields [79]. In this setting, verified benchmark datasets are essential; however, producing a high quality benchmark dataset is a complex, and effort- and time-consuming process [80]. Most of the available benchmark datasets in the field of kidney cancer are mainly related to the histopathological or radiological evaluation of kidneys (i.e. The 2019 kidney and kidney tumor segmentation challenge [KiTS19], KiTS21, and KMC datasets) [81–83]. According to our knowledge, there is no benchmark datasets related to segmentation of instruments and tissues during minimally invasive partial nephrectomy.”
- The manuscript fails to summarize the challenges such as explainability of models and the state of art for the field of image diagnosis in medical, with the specialization of kidney cancer. Such as following references guide towards use of AI in image diagnosis in medical field.
Varoquaux, G., Cheplygina, V. Machine learning for medical imaging: methodological failures and recommendations for the future. npj Digit. Med. 5, 48 (2022). https://doi.org/10.1038/s41746-022-00592-y
Ziatdinov, M., Ghosh, A., Wong, C.Y.(. et al. AtomAI framework for deep learning analysis of image and spectroscopy data in electron and scanning probe microscopy. Nat Mach Intell 4, 1101–1112 (2022). https://doi.org/10.1038/s42256-022-00555-8
Hou, J., Gao, T. Explainable DCNN based chest X-ray image analysis and classification for COVID-19 pneumonia detection. Sci Rep 11, 16071 (2021). https://doi.org/10.1038/s41598-021-95680-6
Thanks again for your comment. We discussed this issue further in the limitation section as follow:
“Noteworthy, it is not always about the size and quality of the dataset. Gaël Varoquaux and colleagues. effectively shed light on methodological errors in various aspects of clinical imaging within their review [79]. Despite extensive research in this field, the clinical impact remains constrained. The review pinpoints challenges in dataset selection, evaluation methods, and publication incentives, proposing strategies for improvement. It underscores the necessity for procedural and normative changes to unlock the full potential of machine learning in healthcare. The authors stress the tendency for research to be influenced by academic incentives rather than meeting the needs of clinicians and patients. Dataset bias is identified as a significant concern, emphasizing the importance of accurately representative datasets. Robust evaluation methods beyond benchmark performance are called for, along with the adoption of sound statistical practices. The publication process may fall short in promoting clarity, potentially hindering reproducibility and transparency. Researchers are urged to prioritize scientific problem-solving over publication optimization, considering broader impacts beyond benchmarks [79].”
As regards language
The article was revised as regards the language.
Reviewer 3 Report
The article concerns the modern approach to the digital support of the education process of novice surgeons in the field of kidney cancer treatment. Problems related to improving the skills of surgeons are a big problem around the world. The number of experts in a given field is limited, and the number of applicants for the profession is constantly increasing. In this aspect, the availability of high-class specialists is problematic. Beginner surgeons are exposed to the lack of high-quality training courses. In this aspect, the concept of digital support for the teaching process in the form of the use of artificial intelligence appears to be an important tool. The presented systematic literature review is a very good step towards the creation and improvement of AI methods to support the preparation of young doctors, not only surgeons. This approach can also be successfully recommended in other areas of vocational education.
The concept of the work is very good, but the form of presenting the results of the literature review requires significant improvement (Table 1). It is very difficult to read and analyze the presented data. Tabular presentation of the essence of the Authors' work significantly reduces the quality of the presented review. I suggest changing the form of presentation of the obtained research results.
Please pay attention to the style and punctuation of the presented text
Author Response
Reply to reviewers
Thank you very much for taking the time to review this manuscript. Please find the detailed responses below and the corresponding revision changes in the re-submitted files.
Reviewer 3:
The article concerns the modern approach to the digital support of the education process of novice surgeons in the field of kidney cancer treatment. Problems related to improving the skills of surgeons are a big problem around the world. The number of experts in a given field is limited, and the number of applicants for the profession is constantly increasing. In this aspect, the availability of high-class specialists is problematic. Beginner surgeons are exposed to the lack of high-quality training courses. In this aspect, the concept of digital support for the teaching process in the form of the use of artificial intelligence appears to be an important tool. The presented systematic literature review is a very good step towards the creation and improvement of AI methods to support the preparation of young doctors, not only surgeons. This approach can also be successfully recommended in other areas of vocational education.
The concept of the work is very good, but the form of presenting the results of the literature review requires significant improvement (Table 1). It is very difficult to read and analyze the presented data. Tabular presentation of the essence of the Authors' work significantly reduces the quality of the presented review. I suggest changing the form of presentation of the obtained research results.
Thanks for your nice comment. We understand your concern, we simplified the table to make it easier for the reader.
As regards language
The article was revised as regards the language.
Round 2
Reviewer 2 Report
The authors addressed the comments effectively.